# Exploring Model-based Planning with Policy Networks

**Tingwu Wang**[1,2]**& Jimmy Ba**[1,2]
[1] Department of Computer Science, University of Toronto [2] Vector Institute
{tingwuwang,jba}@cs.toronto.edu

## Abstract

Model-based reinforcement learning (MBRL) with model-predictive control or online planning has shown great potential for locomotion control tasks in both sample efficiency and asymptotic performance. Despite the successes, the existing planning methods search from candidate sequences randomly generated in the action space, which is inefficient in complex high-dimensional environments. In this paper, we propose a novel MBRL algorithm, model-based policy planning (POPLIN), that combines policy networks with online planning. More specifically, we formulate action planning at each time-step as an optimization problem using neural networks. We experiment with both optimization w.r.t. the action sequences initialized from the policy network, and also online optimization directly w.r.t. the parameters of the policy network. We show that in the MuJoCo benchmarking environments, POPLIN is about 3x more sample efficient than the previously state-of-the-art algorithms, such as PETS, TD3 and SAC. To explain the effectiveness of our algorithm, we show that the optimization surface in parameter space is smoother than in action space. Further more, we found the distilled policy network can be effectively applied without the expansive model predictive control during test time for some environments such as Cheetah. Code is released here[1].

## 1 Introduction

A model-based reinforcement learning (MBRL) agent learns its internal model of the world, i.e. the dynamics, from repeated interactions with the environment. With the learnt dynamics, a MBRL agent can for example perform online planning, interact with imaginary data, or optimize the controller through dynamics, which provides significantly better sample efficiency (Deisenroth & Rasmussen, 2011; Sutton, 1990; Levine & Abbeel, 2014; Levine & Koltun, 2013). However, MBRL algorithms generally do not scale well with the increasing complexity of the reinforcement learning (RL) tasks in practice. And modelling errors in dynamics that accumulate with time-steps greatly limit the applications of MBRL algorithms. As a result, many latest progresses in RL has been made with model-free reinforcement learning (MFRL) algorithms that are capable of solving complex tasks at the cost of large number of samples (Schulman et al., 2017; Heess et al., 2017; Schulman et al., 2015; Mnih et al., 2013; Lillicrap et al., 2015; Haarnoja et al., 2018).

With the success of deep learning, a few recent works have proposed to learn neural network-based dynamics models for MBRL. Among them, random shooting algorithms (RS), which uses model-predictive control (MPC), is shown to have good robustness and scalability (Richards, 2005). In shooting algorithms, the agent randomly generates action sequences, use the dynamics to predict the future states, and choose the first action from the sequence with the best expected reward. However, RS usually has worse asymptotic performance than model-free controllers (Nagabandi et al., 2017), and the authors of the the PETS algorithm (Chua et al., 2018) suggest that the performance of RS is directly affected by the quality of the learnt dynamics. They propose a probabilistic ensemble to capture model uncertainty, which enables PETS algorithm to achieve both better sample efficiency and better asymptotic performance than state-of-the-art model-free controllers in environments such as Cheetah. However, PETS is not as effective on environments with higher dimensionality.

---

[1]https://github.com/WilsonWangTHU/POPLIN.

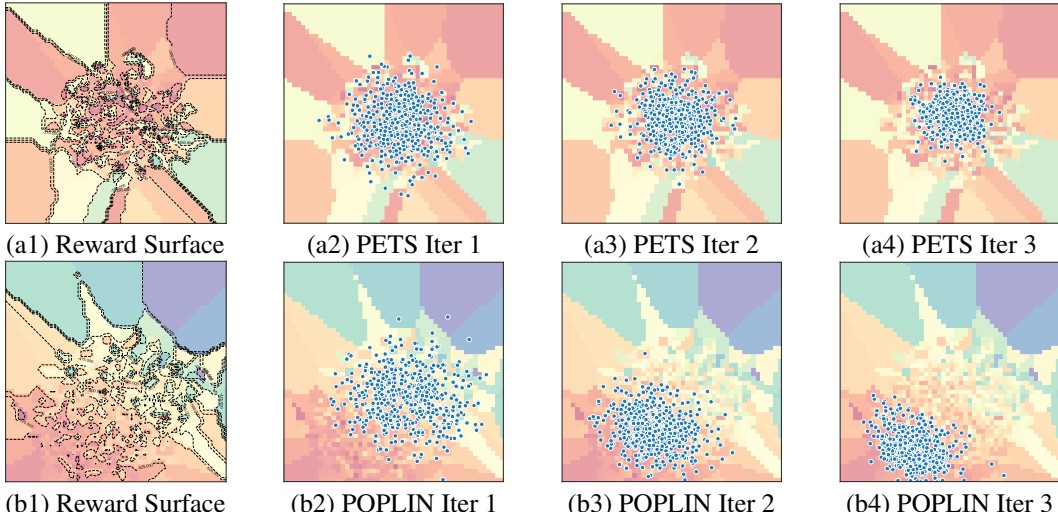

Figure 1: We transform each planned candidate action trajectory with PCA into a 2D blue scatter. The top and bottom figures are respectively the visualization of PETS (Chua et al., 2018) and our algorithm. The red area has higher reward. From left to right, we show how candidate trajectories are updated, across different planning iterations within one time-step. As we can see, while both reward surface is not smooth with respect to action trajectory. POPLIN, using policy networks, has much better search efficiency, while PETS is stuck around its initialization. The details are in section 5.3.

In this paper, we explore MBRL algorithms from a different perspective, where we treat the planning at each time-step as an optimization problem. Random search in action space, as what is being done in state-of-the-art MBRL algorithms such as PETS, is insufficient for more complex environments. On the one hand, we are inspired by the success of AlphaGo (Silver et al., 2016; 2017), where a policy network is used to generate proposals for the Monte-Carlo tree search. On the other hand, we are inspired by the recent research into understanding deep neural networks (Nguyen & Hein, 2017; Li et al., 2018; Soudry & Hoffer, 2017). Deep neural networks, frequently observed in practices, is much less likely to get stuck in sub-optimal points. In Figure 1, we apply principal component analysis (PCA) on the action sequences generated in each planning iteration within one time-step. The reward surface of the action space is not smooth and prone to local-minimas. We argue that optimization in the policy network's parameter space will be more efficient. Furthermore, we note that the state-of-the-art MBRL algorithm with MPC cannot be applied real-time. We therefore experiment with different policy network distillation schemes for fast control without MPC. To sum up, the contribution of this paper is three-fold:

- We apply policy networks to generate proposals for MPC in high dimensional locomotion control problems with unknown dynamics.

- We formulate planning as optimization with neural networks, and propose policy planning in parameter space, which obtain state-of-the-art performance on current bench-marking environments, being about 3x more sample efficient than the previous state-of-the-art algorithm, such as PETS (Chua et al., 2018), TD3 (Fujimoto et al., 2018) and SAC (Haarnoja et al., 2018).

- We also explore policy network distillation from the planned trajectories. We found the distilled policy network alone achieves high performance on environments like Cheetah without the expansive online planning.

## 2 RELATED WORK

Model-based reinforcement learning (MBRL) has been long studied. Dyna (Sutton, 1990; 1991) algorithm alternately performs sampling in the real environments and optimize the controllers on the learned model of the environments. Other pioneering work includes PILCO (Deisenroth & Rasmussen, 2011), where the authors model the dynamics using Gaussian Process and directly optimize the surrogate expected reward. Effective as it is to solve simple environments, PILCO

heavily suffers the curse of dimensionality. In (Levine & Abbeel, 2014; Levine & Koltun, 2013; Levine et al., 2016; Chebotar et al., 2017; Zhang et al., 2018), the authors propose guided policy search (GPS). GPS uses iLQG (Li & Todorov, 2004; Todorov & Li, 2005; Tassa et al., 2012) as the local controller, and distill the knowledge into a policy neural network. In SVG (Heess et al., 2015), the authors uses stochastic value gradient so that the stochastic policy network can be optimized by back-propagation with off-policy data. Recently with the progress of model-free algorithms such as TRPO and PPO (Schulman et al., 2015; 2017), Kurutach et al. (2018); Luo et al. (2019) propose modern variants of Dyna, where TRPO (Schulman et al., 2015) is used to optimize the policy network using data generated by the learnt dynamics. Concurrent to this work, Janner et al. (2019) further use SAC (Haarnoja et al., 2018) to train the policy network, and gets state-of-the-art performance on many tasks. At the same time, random shooting methods proposed by Nagabandi et al. (2017); Chua et al. (2018) have shown its robustness and effectiveness on benchmarking environments. PETS algorithm (Chua et al., 2018) is considered by many to be the state-of-the-art shooting algorithm, which we discuss in detail in section 3. Dynamics is also used to obtain better value estimation to speed up training (Gu et al., 2016; Feinberg et al., 2018; Buckman et al., 2018). Latent dynamics models using VAE (Kingma & Welling, 2013) are commonly used to solve problems with image input (Ha & Schmidhuber, 2018a;b; Hafner et al., 2018; Kaiser et al., 2019).

# 3 BACKGROUND

## 3.1 REINFORCEMENT LEARNING

In reinforcement learning, the problem of solving the given task is formulated as a infinite-horizon discounted Markov decision process. For the agent, we denote the action space and state space respectively as $\mathcal{A}$ and $\mathcal{S}$. We also denote the reward function and transition function as $r(s_t, a_t)$ and $f(s_{t+1}|s_t, a_t)$, where $s_t \in \mathcal{S}$ and $a_t \in \mathcal{A}$ are the state and action at time-step $t$. The reward function is assumed known to the agent in this work. The agent maximizes its expected total reward $J(\pi) = \mathbb{E}_\pi[\sum_{t=0}^{\infty} r(s_t, a_t)]$ with respect to the agent's controller $\pi$.

## 3.2 RANDOM SHOOTING ALGORITHM AND PETS

Our proposed algorithm is based on the random shooting algorithm (Richards, 2005). In random shooting algorithms (Nagabandi et al., 2017; Chua et al., 2018), a data-set of $\mathcal{D} = \{(s_t, a_t, s_{t+1})\}$ is collected from previously generated real trajectories. The agent learns an ensemble of neural networks denoted as $f_\phi(s_{t+1}|s_t, a_t)$, with the parameters of the neural networks denoted as $\phi$. In planning, the agent randomly generates a population of $K$ candidate action sequences. Each action sequence, denoted as $\mathbf{a} = \{a_0, ..., a_\tau\}$, contains the control signals at every time-steps within the planning horizon $\tau$. The action sequence with the best expected reward given the current dynamics network $f_\phi(s_{t+1}|s_t, a_t)$ is chosen. RS, as a model-predictive control algorithm, only executes the first action signal and re-plan at time-step. In PETS (Chua et al., 2018), the authors further use cross entropy method (CEM) (De Boer et al., 2005; Botev et al., 2013) to re-samples sequences near the best sequences from the last CEM iteration.

# 4 MODEL-BASED POLICY PLANNING

In this section, we describe two variants of POPLIN: model-based policy planning in action space (**POPLIN-A**) and model-based policy planning in parameter space (**POPLIN-P**). Following the notations in section 3.2, we define the expected planning reward function at time-step $i$ as follows:

---

**Algorithm 1** General POPLIN Framework

1: **while** Training iterations not Finished **do**
2:     **for** $i^{th}$ time-step of the agent **do**
3:         CEM planning as in section 4.1, 4.2
4:         Execute the first action from CEM.
5:     **end for**
6:     Dynamics update and policy distillation.
7: **end while**

---

$$\mathcal{R}(s_i, \mathbf{a}_i) = \mathbb{E}\left[\sum_{t=i}^{i+\tau} r(s_t, a_t)\right], \text{ where } s_{t+1} \sim f_\phi(s_{t+1}|s_t, a_t). \tag{1}$$

The action sequence $\mathbf{a}_i = \{a_i, a_{i+1}, ..., a_{i+\tau}\}$ is generated by the policy search module, as later described in Section 4.1 and 4.2. The expectation of predicted trajectories $\{s_i, s_{i+1}, ..., s_{i+\tau}\}$ is

estimated by creating $P$ particles from the current state. The dynamics model $f_\phi^{k,t}(s_{t+1}|s_t, a_t)$ used by $k^{th}$ particle at time-step $t$ is sampled from deterministic or probabilistic ensemble models. To better illustrate, throughout the paper we denote this dynamics as a fixed deterministic model, i.e. $f_\phi^{k,t} \equiv f_\phi$. In practice the dynamics uses probabilistic ensemble models, which requires some trivial modifications to the math and we refer readers to PETS Chua et al. (2018) for details.

## 4.1 MODEL-BASED POLICY PLANNING IN ACTION SPACE

In model-based policy planning in action space (POPLIN-A), we use a policy network to generate good initial action distribution. We denote the policy network as $\pi(s_t)$. Once the policy network proposes sequences of actions on the expected trajectories, we add Gaussian noise to the candidate actions and use CEM to fine-tune the mean and standard deviation of the noise distribution.

Similar to defining $\mathbf{a}_i = \{a_i, a_{i+1}, ..., a_{i+\tau}\}$, we denote the noise sequence at time-step $t$ with horizon $\tau$ as $\boldsymbol{\delta}_i = \{\delta_i, \delta_{i+1}, ..., \delta_{i+\tau}\}$. We initialize the noise distribution as a Gaussian distribution with mean $\mu_0 = \mathbf{0}$ and covariance $\Sigma_0 = \sigma_0^2 \boldsymbol{I}$, where $\sigma_0^2$ is the initial noise variance. In each CEM iteration, we first sort out the sequences with the top $\xi + 1$ expected planning reward, whose noise sequences are denoted as $\{\boldsymbol{\delta}_i^0, \boldsymbol{\delta}_i^1, ..., \boldsymbol{\delta}_i^\xi\}$. Then we estimate the noise distribution of the elite candidates, i. e.,

$$\Sigma' \leftarrow \text{Cov}(\{\boldsymbol{\delta}_i^0, \boldsymbol{\delta}_i^1, ..., \boldsymbol{\delta}_i^\xi\}), \ \mu' \leftarrow \text{Mean}(\{\boldsymbol{\delta}_i^0, \boldsymbol{\delta}_i^1, ..., \boldsymbol{\delta}_i^\xi\}). \tag{2}$$

The elite distribution $(\mu', \Sigma')$ in CEM algorithm is used to update the candidate noise distribution as $\mu = (1 - \alpha)\mu + \alpha\mu'$, $\Sigma = (1 - \alpha)\Sigma + \alpha\Sigma'$. For every time-step, several CEM iterations are performed by candidate re-sampling and noise distribution updating. We provide detailed algorithm boxes in appendix A.1. We consider the following two schemes to add action noise.

**POPLIN-A-Init:** In this planning schemes, we use the policy network only to propose the initialization of the action sequences. When planning at time-step $i$ with observed state $s_i$, we first obtain the initial reference action sequences, denoted as $\hat{\mathbf{a}}_i = \{\hat{a}_i, \hat{a}_{i+1}, ..., \hat{a}_{i+\tau}\}$, by running the initial forward pass with policy network. At each planning time-step $t$, where $i \leq t \leq i + \tau$, we have $\hat{a}_t = \pi(\hat{s}_t)$, where $\hat{s}_t = f_\phi(\hat{s}_{t-1}, a_{t-1})$, $\hat{s}_i = s_i$ The expected reward given search noise $\boldsymbol{\delta}_i$ will be:

$$\mathcal{R}(s_i, \boldsymbol{\delta}_i) = \mathbb{E}\left[\sum_{t=i}^{i+\tau} r(s_t, \hat{a}_t + \delta_t)\right], \text{ where } s_{t+1} = f_\phi(s_{t+1}|s_t, \hat{a}_t + \delta_t). \tag{3}$$

**POPLIN-A-Replan:** POPLIN-A-Replan is a more aggressive planning schemes, which always re-plans the controller according the changed trajectory given the current noise distribution. If we had the perfect dynamics network and the policy network, then we expect re-planning to achieve faster convergence the optimal action distribution. But it increases the risk of divergent behaviors. In this case, the expected reward for each trajectory is

$$\mathcal{R}(s_i, \boldsymbol{\delta}_i) = \mathbb{E}\left[\sum_{t=i}^{i+\tau} r(s_t, \pi(s_t) + \delta_t)\right], \text{ where } s_{t+1} = f_\phi(s_{t+1}|s_t, \pi(s_t) + \delta_t). \tag{4}$$

## 4.2 MODEL-BASED POLICY PLANNING IN PARAMETER SPACE

While planning in the action space is a natural extension of the original PETS algorithm, we found it provides little performance improvement in complex environments. One potential reason is that POPLIN-A still performs CEM searching in action sequence space, where the conditions of convergence for CEM is usually not met. Let's assume that a robot arm needs to either go left or right to get past the obstacle in the middle. In CEM planning in the action space, the theoretic distribution mean is always going straight, which fails to model the bi-modal action distribution.

Indeed, planning in action space is a non-convex optimization whose surface has lots of holes and peaks. Recently, much research progress has been made in understanding why deep neural networks are much less likely to get stuck in sub-optimal points Nguyen & Hein (2017); Li et al. (2018); Soudry & Hoffer (2017). And we believe that planning in parameter space is essentially using deeper neural networks. Therefore, we propose model-based policy planning in parameter space (POPLIN-P).

Instead of adding noise in the action space, POPLIN-P adds noise in the parameter space of the policy network. We denote the parameter vector of policy network as $\theta$, and the parameter noise sequence starting from time-step $i$ as $\boldsymbol{\omega}_i = \{\omega_i, \omega_{i+1}, ..., \omega_{i+\tau}\}$. The expected reward function is now

$$\mathcal{R}(s_i, \boldsymbol{\omega}_i) = \mathbb{E}\left[\sum_{t=i}^{i+\tau} r\left(s_t, \pi_{\theta+\omega_t}(s_t)\right)\right], \text{ where } s_{t+1} = f_\phi(s_{t+1}|s_t, \pi_{\theta+\omega_t}(s_t)). \quad (5)$$

Similarly, we update the CEM distribution towards the following elite distribution:

$$\Sigma' \leftarrow \text{Cov}(\{\boldsymbol{\omega}_i^0, \boldsymbol{\omega}_i^1, ..., \boldsymbol{\omega}_i^\xi\}), \ \mu' \leftarrow \text{Mean}(\{\boldsymbol{\omega}_i^0, \boldsymbol{\omega}_i^1, ..., \boldsymbol{\omega}_i^\xi\}). \quad (6)$$

We can force the policy network noise within the sequence to be consistent, i.e. $\omega_i = \omega_{i+1} = ... = \omega_{i+\tau}$, which we name as **POPLIN-P-Uni**. This reduces the size of the flattened noise vector from $(\tau + 1)|\theta|$ to $|\theta|$, and is more consistent in policy behaviors. The noise can also be separate for each time-step, which we name as **POPLIN-P-Sep**. We benchmark both schemes in section 5.4.

**Equivalence to re-parameterized stochastic policy:** Stochastic policy network encourages exploration, and increases the robustness against the impact of compounded model errors. POPLIN-P, which inserts exogenous noise into the parameter space, can be regarded as a re-parameterized stochastic policy network, which natural combines stochastic policy network with planning.

## 4.3 Model-predictive Control and Policy Control

MBRL with online re-planning or model-predictive control (MPC) is effective, but at the same time time-consuming. Many previous attempts have tried to distill the planned trajectories into a policy network Levine & Abbeel (2014); Levine & Koltun (2013); Chebotar et al. (2017); Zhang et al. (2018), and control only with policy network. In this paper, we define two settings of using POPLIN: **MPC Control** and **Policy Control**. In MPC control, the agent uses policy network during the online planning and only execute the first action. In policy control, the agent directly executes the signal produced by the policy network given current observation, just like how policy network is used in MFRL algorithms. We show both performance of POPLIN in this paper.

## 4.4 Policy Distillation Schemes

The agents iterate between interacting with the environments, and distilling the knowledge from planning trajectory into a policy network. We consider several policy distillation schemes here, and discuss their effectiveness in the later experimental section.

**Behavior cloning (BC)**: BC can be applied to POPLIN-A and POPLIN-P, by minimizing the squared L2 loss as Equation 7. $\mathcal{D}$ is the collection of observation and planned action from real environment. When applying BC to POPLIN-P, we fix parameter noise of the network to be zeros.

$$\min_\theta \mathbb{E}_{s,\,a \in \mathcal{D}}||\pi_\theta(s) - a||^2. \quad (7)$$

**Generative adversarial network training (GAN)** Goodfellow et al. (2014): GAN can be applied to POPLIN-P. We consider the following fact. During MPC control, the agent only needs to cover the best action sequence in its action sequence distribution. Therefore, instead of point-to-point supervised training such as BC, we can train the policy network using GAN:

$$\min_{\pi_\theta} \max_\psi \mathbb{E}_{s,\,a \in \mathcal{D}} \log(D_\psi(s, a)) + \mathbb{E}_{s \in \mathcal{D},\,z \sim \mathcal{N}(\mathbf{0}, \sigma_0 \boldsymbol{I})} \log(1 - D_\psi(s, \pi_{\theta+z}(s))), \quad (8)$$

where a discriminator $D$ parameterized by $\psi$ is used, and we sample the random noise $z$ from the initial CEM distribution $\mathcal{N}(\mathbf{0}, \sigma_0 \boldsymbol{I})$.

**Setting parameter average (AVG)**: AVG is also applicable to POPLIN-P. During interaction with real environment, we also record the optimized parameter noise in to the data-set, i. e. $\mathcal{D} = \{(s, \omega)\}$. And we sacrifice the effectiveness of the policy control and only use policy network as a good search initialization. The new parameter is updated as $\theta = \theta + 1/|\mathcal{D}| \sum_{\omega \in \mathcal{D}} \omega$.

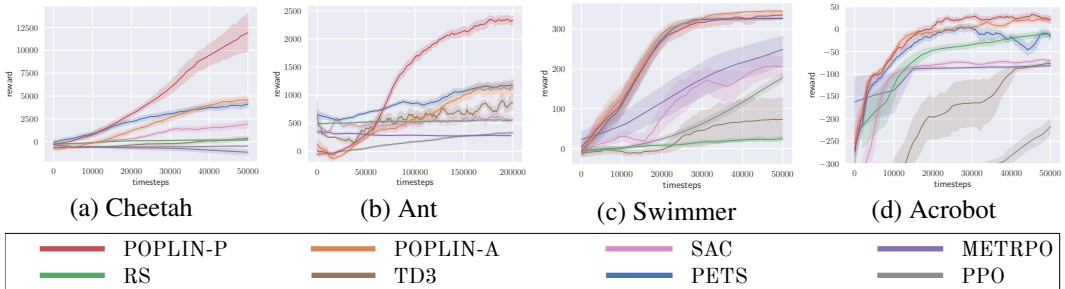

Figure 2: Performance curves on different bench-marking environments. 4 random seeds are run for each environment. The full figures of all 12 MuJoCo environments are summarized in appendix 8.

| | Cheetah | Ant | Hopper | Swimmer | Cheetah-v0 | Walker2d |
|---|---|---|---|---|---|---|
| POPLIN-P (ours) | **12227.9 ± 5652.8** | **2330.1 ± 320.9** | **2055.2 ± 613.8** | 334.4 ± 34.2 | **4235.0 ± 1133.0** | **597.0 ± 478.8** |
| POPLIN-A (ours) | 4651.1 ± 1088.5 | 1148.4 ± 438.3 | 202.5 ± 962.5 | **344.9 ± 7.1** | 1562.8 ± 1136.7 | -105.0 ± 249.8 |
| PETS (Chua et al., 2018) | 4204.5 ± 789.0 | 1165.5 ± 226.9 | 114.9 ± 621.0 | 326.2 ± 12.6 | 2288.4 ± 1019.0 | 282.5 ± 501.6 |
| METRPO (Kurutach et al., 2018) | -744.8 ± 707.1 | 282.2 ± 18.0 | 1272.5 ± 500.9 | 225.5 ± 104.6 | 2283.7 ± 900.4 | -1609.3 ± 657.5 |
| TD3 (Fujimoto et al., 2018) | 218.9 ± 593.3 | 870.1 ± 283.8 | 1816.6 ± 994.8 | 72.1 ± 130.9 | 3015.7 ± 969.8 | -516.4 ± 812.2 |
| SAC (Haarnoja et al., 2018) | 1745.9 ± 839.2 | 548.1 ± 146.6 | 788.3 ± 738.2 | 204.6 ± 69.3 | 3459.8 ± 1326.6 | 164.5 ± 1318.6 |
| Training Time-step | 50000 | 200000 | 200000 | 50000 | 200000 | 200000 |

| | Reacher3D | Pusher | Pendulum | InvertedPendulum | Acrobot | Cartpole |
|---|---|---|---|---|---|---|
| POPLIN-P (ours) | **-29.0 ± 25.2** | **-55.8 ± 23.1** | 167.9 ± 45.9 | **-0.0 ± 0.0** | **23.2 ± 27.2** | **200.8 ± 0.3** |
| POPLIN-A (ours) | **-27.7 ± 25.2** | -56.0 ± 24.3 | **178.3 ± 19.3** | **-0.0 ± 0.0** | 20.5 ± 20.1 | **200.6 ± 1.3** |
| PETS (Chua et al., 2018) | -47.7 ± 43.6 | **-52.7 ± 23.5** | 155.7 ± 79.3 | -29.5 ± 37.8 | -18.4 ± 46.3 | 199.6 ± 4.6 |
| METRPO (Kurutach et al., 2018) | -43.5 ± 3.7 | -98.5 ± 12.6 | 174.8 ± 6.2 | -29.3 ± 29.5 | -78.7 ± 5.0 | 138.5 ± 63.2 |
| TD3 (Fujimoto et al., 2018) | -331.6 ± 134.6 | -216.4 ± 39.6 | 168.6 ± 12.7 | -102.9 ± 101.0 | -76.5 ± 10.2 | -409.2 ± 928.8 |
| SAC (Haarnoja et al., 2018) | -161.6 ± 43.7 | -227.6 ± 42.2 | 159.5 ± 12.1 | -0.2 ± 0.1 | -69.4 ± 7.0 | 195.5 ± 8.7 |
| Training Time-step | 50000 | 50000 | 50000 | 50000 | 50000 | 50000 |

Table 1: The training time-step varies from 50,000 to 200,000 depending on the difficulty of the tasks. The performance is averaged across four random seeds with the last 3 episodes.

## 5 EXPERIMENTS

In section 5.1, we compare POPLIN with existing algorithms. We also show the policy control performance of POPLIN with different training methods in section 5.2. In section 5.3, we provide explanations and analysis for the effectiveness of our proposed algorithms by exploring and visualizing the planner's reward optimization surface. In section 5.4, we study the sensitivity of our algorithms with respect to hyper-parameters, and show the performance of different algorithm variants.

### 5.1 MUJOCO BENCHMARKING PERFORMANCE

In this section, we compare POPLIN with existing reinforcement learning algorithms including PETS (Chua et al., 2018), GPS (Levine et al., 2016), RS (Richards, 2005), MBMF (Nagabandi et al., 2017), TD3 (Fujimoto et al., 2018) METRPO (Kurutach et al., 2018), PPO (Schulman et al., 2017; Heess et al., 2017), TRPO (Schulman et al., 2015) and SAC (Haarnoja et al., 2018), which includes the most recent progress of both model-free and model-based algorithms. We examine the algorithms with 12 environments, which is a wide collection of environments from OpenAI Gym (Brockman et al., 2016) and the environments proposed in PETS (Chua et al., 2018), which are summarized in appendix A.2. Due to the page limit and to better visualize the results, we put the complete figures and tables in appendix A.3. And in Figure 2 and Table 1, we show the performance of our algorithms and the best performing baselines. The hyper-parameter search is summarized in appendix A.3.1.

As shown in Table 1, POPLIN achieves state-of-the-art performance in almost all environments, solving most of the them with 200,000 or 50,000 time-steps, instead of 1 million time-steps commonly used in MFRL algorithms. POPLIN-A (POPLIN-A-BC-Replan) has the best performance in simpler environments such as Pendulum, Cart-pole, Swimmer. But on complex environments such as Ant, Cheetah or Hopper, POPLIN-A does not have obvious performance gain compared with PETS. POPLIN-P (POPLIN-P-Sep-AVG) on the other hand, has consistent and stable performance among different environments. POPLIN-P is significantly better than all other algorithms in complex environments such as Ant and Cheetah. However, like other model-based algorithms, POPLIN cannot solve environments such as Walker and Humanoid. the performance of POPLIN plateaus

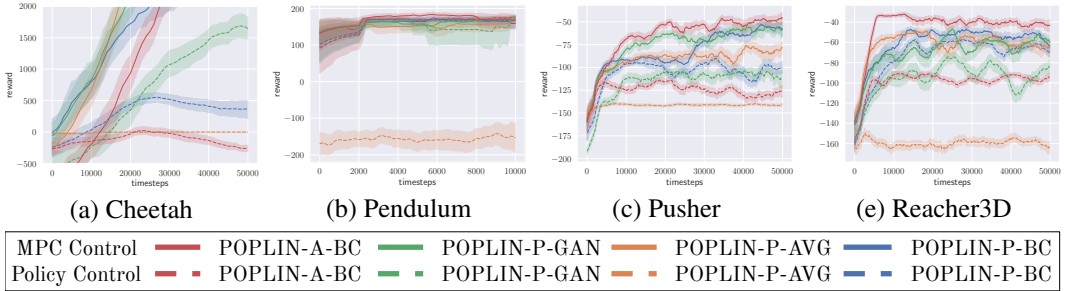

Figure 3: The MPC control and policy control performance of the proposed POPLIN-A, and POPLIN-P with its three training schemes, which are namely behavior cloning (BC), generative adversarial network training (GAN) and setting parameter average (Avg).

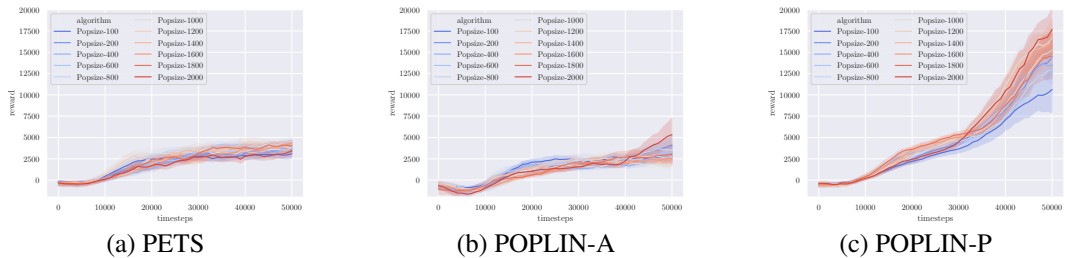

Figure 4: The performance of PETS, POPLIN-A, POPLIN-P using different population size of candidates on Cheetah. The variance of the candidates trajectory $\sigma$ in POPLIN-P is set to 0.1.

quickly. Gradually model-free algorithms will have better asymptotic performance. We view this as a bottleneck of our algorithms and leave it to future research.

## 5.2 POLICY CONTROL PERFORMANCE

In this section, we show the performance of POPLIN without MPC. To be more specific, we show the performance with the Cheetah, Pendulum, Pusher and Reacher3D, as shown in Figure 3, and we refer readers to appendix A.4 for the full results.

We note that policy control is not always successful, and in environments such as Ant and Walker2D, the performance is almost random. In simple environments such as Pusher and Reacher3D, POPLIN-A has the best MPC performance, but has worse policy control performance compared with POPLIN-P-BC and POPLIN-P-GAN. At the same time, both POPLIN-P-BC and POPLIN-P-GAN are able to efficiently distill the knowledge from planned trajectory. Which one of POPLIN-P-BC and POPLIN-P-GAN is better depends on the environment tested, and they can be used interchangeably. This indicates that POPLIN-A, which uses a deterministic policy network, is more prone to distillation collapse than POPLIN-P, which can be interpreted as using a stochastic policy network with reparameterization trick. POPLIN-P-Avg, which only use policy network as optimization initialization has good MPC performance, but sacrifices the policy control performance. In general, the performance of policy control lags behind MPC control.

## 5.3 SEARCH EFFECTIVENESS AND REWARD SURFACE

In this section, we explore the reasons for the effectiveness of POPLIN. In Figure 4, we show the performance of PETS, POPLIN-A and POPLIN-P with different population sizes. As we can see, PETS and POPLIN-A, which are the two algorithms that add search noise in the action space, cannot increase their performance by having bigger population size. However, POPLIN-P is able to efficiently increase performance with bigger population size. We then visualize the candidates in their reward or optimization surface in Figure 1. We use PCA (principal component analysis) to transform the action sequences into 2D features. As we can see, the reward surface is not smooth, with lots of local-minima and local-maxima islands. The CEM distribution of PETS algorithm is almost fixed across iterations on this surface, even if there are potentially higher reward regions. POPLIN is able

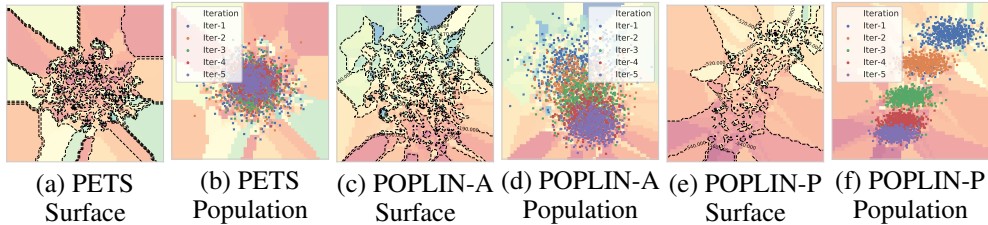

| (a) PETS Surface | (b) PETS Population | (c) POPLIN-A Surface | (d) POPLIN-A Population | (e) POPLIN-P Surface | (f) POPLIN-P Population |

Figure 5: The reward optimization surface in the solution space. The expected reward is higher from color blue to color red. We visualize candidates using different colors as defined in the legend. The full results can be seen in appendix A.7.

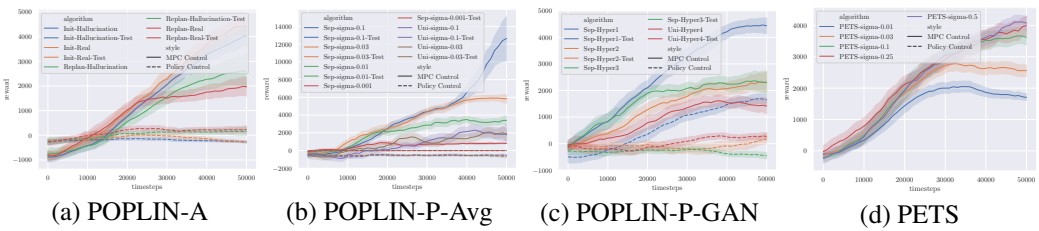

| (a) POPLIN-A | (b) POPLIN-P-Avg | (c) POPLIN-P-GAN | (d) PETS |

Figure 7: The ablation study of of POPLIN-A, POPLIN-P-BC, POPLIN-P-Avg, POPLIN-P-GAN.

to efficiently search through the jagged reward surface, from the low-reward center to the high reward left-down corner. To further understand why POPLIN is much better at searching through the reward surface, we then plot the figures in the solution space in Figure 5. More specifically, we now perform PCA on the policy parameters for POPLIN-P. As we can see in Figure 5 (c), the reward surface in parameter space is much smoother than the reward surface in action space, which are shown in Figure 5 (a), (b). POPLIN-P can efficiently search through the smoother reward surface in parameter space.

In Figure 6, we also visualize the actions distribution in one episode taken by PETS, POPLIN-A and POPLIN-P using policy networks of different number of hidden layers. We again use PCA to project the actions into 2D feature space. As we can see, POPLIN-P shows a clear pattern of being more multi-modal with the use of deeper the network.

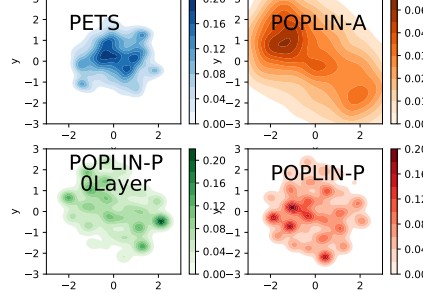

Figure 6: Projected action distribution.

## 5.4 ABLATION STUDY

In this section, we study how sensitive our algorithms are with respect to some of the crucial hyper-parameters, for example, the initial variance of the CEM noise distribution. We also show the performance of different algorithm variants. The full ablation study and performance against different random seeds are included in appendix A.5. In Figure 7 (a), we show the performance of POPLIN-A using different training schemes. We try both training with only the real data samples, which we denote as "Real", and training also with imaginary data the agent plans into the future, which we denote as "Hallucination". In practice, POPLIN-A-Init performs better than POPLIN-A-Replan, which suggests that there can be divergent or overconfident update in POPLIN-A-Replan. And training with or without imaginary does not have big impact on the performance. In Figure7 (b) and (c), we also compare the performance of POPLIN-P-Uni with POPLIN-P-Sep, where we show that POPLIN-P-Sep has much better performance than POPLIN-P-Uni, indicating the search is not efficient enough in the constrained parameter space. For POPLIN-P-Avg, with bigger initial variance of the noise distribution, the agent gets better at planning. However, increasing initial noise variance does not increase the performance of PETS algorithm, as shown in 7 (b), (d). It is worth mentioning that POPLIN-P-GAN is highly sensitive to the entropy penalty we add to the discriminator, with the 3 curves in Figure7 (c) using entropy penalty of 0.003, 0.001 and 0.0001 respectively,

## 6 CONCLUSIONS

In this paper, we explore efficient ways to combine policy networks with model-based planning. We propose POPLIN, which obtains state-of-the-art performance on the MuJoCo benchmarking environments. We study different distillation schemes to provide fast controllers during testing. More importantly, we formulate online planning as optimization using deep neural networks. We believe POPLIN will scale to more complex environments in the future.

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

# A APPENDIX

## A.1 ALGORITHM DIAGRAMS

To better illustrate the algorithm variants of our proposed methods, we summarize them in Algorithm 2, 3, 4.

---

**Algorithm 2** POPLIN-A-Init

---

1: Initialize policy network parameters $\theta$, dynamics network parameters $\phi$, data-set $\mathcal{D}$
2: **while** Training iterations not Finished **do**
3:     **for** $i^{th}$ time-step of the agent **do**                             ▷ Sampling Data
4:         Initialize reference action sequence $\{\hat{a}_i, \hat{a}_{i+1}, ..., \hat{a}_{i+\tau}\}$.       ▷ Using Equation 3
5:         Initialize action-sequence noise distribution. $\mu = \mu_0$, $\Sigma = \sigma_0^2 \boldsymbol{I}$
6:         **for** $j^{th}$ CEM Update **do**                          ▷ CEM Planning
7:             Sample action noise sequences $\{\boldsymbol{\delta}_i\}$ from $\mathcal{N}(\mu, \Sigma)$.
8:             **for** Every candidate $\boldsymbol{\delta}_i$ **do**                ▷ Trajectory Predicting
9:                 for $t = i$ to $i + \tau$, $s_{t+1} = f_\phi(s_{t+1}|s_t, a_t = \hat{a}_t + \delta_t)$
10:                 Evaluate expected reward of this candidate.
11:             **end for**
12:             Fit distribution of the elite candidates as $\mu', \Sigma'$.
13:             Update noise distribution $\mu = (1 - \alpha)\mu + \alpha\mu'$, $\Sigma = (1 - \alpha)\Sigma + \alpha\Sigma'$
14:         **end for**
15:         Execute the first action from the optimal candidate action sequence.
16:     **end for**
17:     Update $\phi$ using data-set $\mathcal{D}$                            ▷ Dynamics Update
18:     Update $\theta$ using data-set $\mathcal{D}$                            ▷ Policy Distillation
19: **end while**

---

---

**Algorithm 3** POPLIN-A-Replan

---

1: Initialize policy network parameters $\theta$, dynamics network parameters $\phi$, data-set $\mathcal{D}$
2: **while** Training iterations not Finished **do**
3:     **for** $i^{th}$ time-step of the agent **do**                             ▷ Sampling Data
4:         Initialize action-sequence noise distribution. $\mu = \mu_0$, $\Sigma = \sigma_0^2 \boldsymbol{I}$
5:         **for** $j^{th}$ CEM Update **do**                          ▷ CEM Planning
6:             Sample action noise sequences $\{\boldsymbol{\delta}_i\}$ from $\mathcal{N}(\mu, \Sigma)$.
7:             **for** Every candidate $\boldsymbol{\delta}_i$ **do**                ▷ Trajectory Predicting
8:                 for $t = i$ to $i + \tau$, $s_{t+1} = f_\phi(s_{t+1}|s_t, a_t = \pi_\theta(s_t) + \delta_t)$
9:                 Evaluate expected reward of this candidate.
10:             **end for**
11:             Fit distribution of the elite candidates as $\mu', \Sigma'$.
12:             Update noise distribution $\mu = (1 - \alpha)\mu + \alpha\mu'$, $\Sigma = (1 - \alpha)\Sigma + \alpha\Sigma'$
13:         **end for**
14:         Execute the first action from the optimal candidate action sequence.
15:     **end for**
16:     Update $\phi$ using data-set $\mathcal{D}$                            ▷ Dynamics Update
17:     Update $\theta$ using data-set $\mathcal{D}$                              ▷ Policy Distillation
18: **end while**

---

## A.2 BENCH-MARKING ENVIRONMENTS

In the original PETS paper Chua et al. (2018), the authors only experiment with 4 environments, which are namely Reacher3D, Pusher, Cartpole and Cheetah. In this paper, we experiment with the 9 more environments based on the standard bench-marking environments from OpenAI Gym Brockman et al. (2016). More specifically, we experiment with InvertedPendulum, Acrobot, Pendulum, Ant, Hopper, Swimmer, Walker2d. We also note that the Cheetah environment in PETS Chua et al. (2018) is different from the standard HalfCheetah-v1 in OpenAI Gym. Therefore we experiment with both versions in our paper, where the Cheetah from PETS is named as "Cheetah", and the HalfCHeetah

---

**Algorithm 4** POPLIN-P

---

1: Initialize policy network parameters $\theta$, dynamics network parameters $\phi$, data-set $\mathcal{D}$
2: **while** Training iterations not Finished **do**
3:     **for** $i^{th}$ time-step of the agent **do**         ▷ Sampling Data
4:         Initialize parameter-sequence noise distribution. $\mu = \mu_0, \Sigma = \sigma_0^2 \boldsymbol{I}$
5:         **for** $j^{th}$ CEM Update **do**         ▷ CEM Planning
6:             Sample parameter noise sequences $\{\boldsymbol{\omega}_i\}$ from $\mathcal{N}(\mu, \Sigma)$.
7:             **for** Every candidate $\boldsymbol{\omega}_i$ **do**         ▷ Trajectory Predicting
8:                 for $t = i$ to $i + \tau$, $s_{t+1} = f_\phi(s_{t+1}|s_t, a_t = \pi_{\theta + \omega_t}(s_t))$
9:                 Evaluate expected reward of this candidate.
10:             **end for**
11:             Fit distribution of the elite candidates as $\mu', \Sigma'$.
12:             Update noise distribution $\mu = (1 - \alpha)\mu + \alpha\mu', \Sigma = (1 - \alpha)\Sigma + \alpha\Sigma'$
13:         **end for**
14:         Execute the first action from the optimal candidate action sequence.
15:     **end for**
16:     Update $\phi$ using data-set $\mathcal{D}$         ▷ Dynamics Update
17:     Update $\theta$ using data-set $\mathcal{D}$         ▷ Policy Distillation
18: **end while**

---

| | Cheetah | Ant | Hopper | Swimmer | Cheetah-v0 | Walker2d | Swimmer-v0 |
|---|---|---|---|---|---|---|---|
| POPLIN-P | 12227.9 ± 5652.8 | 2330.1 ± 320.9 | 2055.2 ± 613.8 | 334.4 ± 34.2 | 4235.0 ± 1133.0 | 597.0 ± 478.8 | 37.1 ± 4.6 |
| POPLIN-A | 4651.1 ± 1088.5 | 1148.4 ± 438.3 | 202.5 ± 962.5 | 344.9 ± 7.1 | 1562.8 ± 1136.7 | -105.0 ± 249.8 | 26.7 ± 13.2 |
| PETS | 4204.5 ± 789.0 | 1165.5 ± 226.9 | 114.9 ± 621.0 | 326.2 ± 12.6 | 2288.4 ± 1019.0 | 282.5 ± 501.6 | 29.7 ± 13.5 |
| RS | 191.1 ± 21.2 | 535.5 ± 37.0 | -2491.5 ± 35.1 | 22.4 ± 9.7 | 421.0 ± 55.2 | -2060.3 ± 228.0 | 26.8 ± 2.3 |
| MBMF | -459.5 ± 62.5 | 134.2 ± 50.4 | -1047.4 ± 1098.7 | 110.7 ± 45.6 | 126.9 ± 72.7 | -2218.1 ± 437.7 | 30.6 ± 4.9 |
| TRPO | -412.4 ± 33.3 | 323.3 ± 24.9 | -2100.1 ± 640.6 | 47.8 ± 11.1 | -12.0 ± 85.5 | -2286.3 ± 373.3 | 26.3 ± 2.6 |
| PPO | -483.0 ± 46.1 | 321.0 ± 51.2 | -103.8 ± 1028.0 | 155.5 ± 14.9 | 17.2 ± 84.4 | -1893.6 ± 234.1 | 24.7 ± 4.0 |
| GPS | 129.4 ± 140.4 | 445.5 ± 212.9 | -768.5 ± 200.9 | -30.9 ± 6.3 | 52.3 ± 41.7 | -1730.8 ± 441.7 | 8.2 ± 10.2 |
| METRPO | -744.8 ± 707.1 | 282.2 ± 18.0 | 1272.5 ± 500.9 | 225.5 ± 104.6 | 2283.7 ± 900.4 | -1609.3 ± 657.5 | 35.4 ± 2.2 |
| TD3 | 218.9 ± 593.3 | 870.1 ± 283.8 | 1816.6 ± 994.8 | 72.1 ± 130.9 | 3015.7 ± 969.8 | -516.4 ± 812.2 | 17.0 ± 12.9 |
| SAC | 1745.9 ± 839.2 | 548.1 ± 146.6 | 788.3 ± 738.2 | 204.6 ± 69.3 | 3459.8 ± 1326.6 | 164.5 ± 1318.6 | 23.0 ± 17.3 |
| Random | -284.2 ± 83.3 | 478.0 ± 47.8 | -2768.0 ± 571.6 | -12.4 ± 12.8 | -312.4 ± 44.2 | -2450.1 ± 406.5 | 2.4 ± 12.0 |
| Time-step | 50000 | 200000 | 200000 | 50000 | 200000 | 200000 | 200000 |

Table 2: Performance of each algorithm on environments based on OpenAI Gym Brockman et al. (2016) MuJoCoTodorov et al. (2012) environments. In the table, we record the performance at 200,000 time-step.

| | Cheetah | Ant | Hopper | Swimmer | Cheetah-v0 | Walker2d | Swimmer-v0 |
|---|---|---|---|---|---|---|---|
| POPLIN-P | 0.944 ± 0.079 | 0.932 ± 0.128 | 0.919 ± 0.112 | 0.936 ± 0.086 | 0.927 ± 0.227 | 0.968 ± 0.15 | 0.928 ± 0.115 |
| POPLIN-A | 0.395 ± 0.057 | 0.459 ± 0.175 | 0.582 ± 0.175 | 0.962 ± 0.018 | 0.393 ± 0.227 | 0.748 ± 0.078 | 0.668 ± 0.33 |
| PETS | 0.363 ± 0.002 | 0.466 ± 0.091 | 0.566 ± 0.113 | 0.916 ± 0.032 | 0.538 ± 0.204 | 0.87 ± 0.157 | 0.743 ± 0.338 |
| RS | 0.072 ± 0.005 | 0.214 ± 0.015 | 0.092 ± 0.006 | 0.156 ± 0.024 | 0.164 ± 0.011 | 0.137 ± 0.071 | 0.67 ± 0.058 |
| MBMF | 0.025 ± 0.002 | 0.054 ± 0.02 | 0.355 ± 0.2 | 0.377 ± 0.114 | 0.105 ± 0.015 | 0.088 ± 0.137 | 0.765 ± 0.123 |
| TRPO | 0.028 ± 0.003 | 0.129 ± 0.01 | 0.164 ± 0.116 | 0.22 ± 0.028 | 0.078 ± 0.017 | 0.067 ± 0.117 | 0.658 ± 0.065 |
| PPO | 0.023 ± 0.01 | 0.128 ± 0.02 | 0.527 ± 0.187 | 0.489 ± 0.037 | 0.083 ± 0.017 | 0.19 ± 0.073 | 0.618 ± 0.1 |
| GPS | 0.067 ± 0.051 | 0.178 ± 0.085 | 0.406 ± 0.037 | 0.023 ± 0.016 | 0.09 ± 0.008 | 0.24 ± 0.138 | 0.205 ± 0.255 |
| METRPO | 0.004 ± 0.043 | 0.113 ± 0.007 | 0.777 ± 0.091 | 0.664 ± 0.262 | 0.537 ± 0.18 | 0.278 ± 0.205 | 0.885 ± 0.055 |
| TD3 | 0.074 ± 0.061 | 0.348 ± 0.114 | 0.876 ± 0.181 | 0.28 ± 0.327 | 0.683 ± 0.194 | 0.62 ± 0.254 | 0.425 ± 0.323 |
| SAC | 0.184 ± 0.006 | 0.219 ± 0.059 | 0.689 ± 0.134 | 0.612 ± 0.173 | 0.772 ± 0.265 | 0.833 ± 0.412 | 0.575 ± 0.433 |
| Random | 0.037 ± 0 | 0.191 ± 0.019 | 0.042 ± 0.104 | 0.069 ± 0.032 | 0.018 ± 0.009 | 0.016 ± 0.127 | 0.06 ± 0.3 |
| max, min | 13000, -800 | 2500, 0 | 2500, -3000 | 360, -40 | -400, 4600 | 700, -2500 | 40, 0 |

Table 3: The normalized performance of Table 2.

from OpenAI Gym is named as "Cheetah-v0". Empirically, Cheetah is much easier to solve than Cheetah-v0, as show in Table 2 and Table 4. We also include two swimmer, which we name as Swimmer and Swimmer-v0, which we explain in section A.2.1.

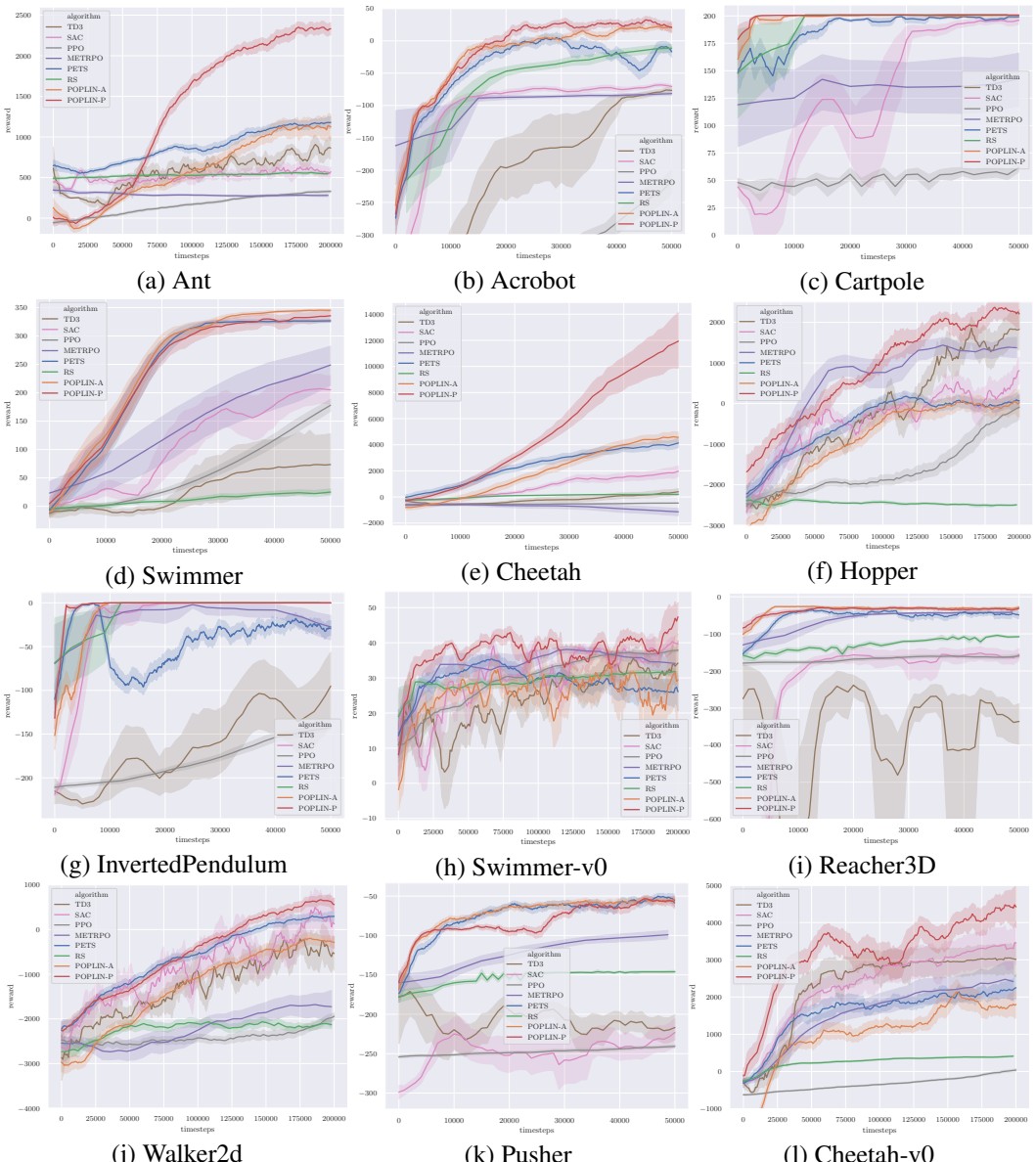

Figure 8: Full Performance of POPLIN-P, POPLIN-A and other state-of-the-art algorithms on 12 different bench-marking environments. In the figure, we include baselines such as TD3, SAC, PPO, METRPO, PETS, RS and our proposed algorithm.

### A.2.1  FIXING THE SWIMMER ENVIRONMENTS

We also notice that after an update in the Gym environments, the swimmer became unsolvable for almost all algorithms. The reward threshold for solving is around 340 for the original swimmer, but almost all algorithms, including the results shown in many published papers Schulman et al. (2017), will be stuck at the 130 reward local-minima. We note that this is due the fact that the velocity sensor is on the neck of the swimmer, making swimmer extremely prone to this performance local-minimum. We provide a fixed swimmer, which we name as Swimmer, by moving the sensor from the neck to the head. We believe this modification is necessary to test the effectiveness of the algorithms.

| | Reacher3D | Pusher | Pendulum | InvertedPendulum | Acrobot | Cartpole |
|---|---|---|---|---|---|---|
| POPLIN-P | -29.0 ± 25.2 | -55.8 ± 23.1 | 167.9 ± 45.9 | -0.0 ± 0.0 | 23.2 ± 27.2 | 200.8 ± 0.3 |
| POPLIN-A | -27.7 ± 25.2 | -56.0 ± 24.3 | 178.3 ± 19.3 | -0.0 ± 0.0 | 20.5 ± 20.1 | 200.6 ± 1.3 |
| PETS | -47.7 ± 43.6 | -52.7 ± 23.5 | 155.7 ± 79.3 | -29.5 ± 37.8 | -18.4 ± 46.3 | 199.6 ± 4.6 |
| RS | -107.6 ± 5.2 | -146.4 ± 3.2 | 161.2 ± 11.5 | -0.0 ± 0.0 | -12.5 ± 14.3 | 201.0 ± 0.0 |
| MBMF | -168.6 ± 23.2 | -285.8 ± 15.2 | 163.7 ± 15.2 | -202.3 ± 17.0 | -146.8 ± 29.9 | 22.5 ± 67.7 |
| TRPO | -176.5 ± 24.3 | -235.5 ± 6.2 | 158.7 ± 9.1 | -134.6 ± 6.9 | -291.2 ± 6.7 | 46.3 ± 6.0 |
| PPO | -162.2 ± 15.7 | -243.2 ± 6.9 | 160.9 ± 12.5 | -137.3 ± 12.4 | -205.4 ± 51.5 | 68.8 ± 4.9 |
| GPS | -552.8 ± 577.7 | -151.2 ± 1.3 | 164.3 ± 4.1 | -14.7 ± 20.7 | -214.3 ± 15.3 | -18.7 ± 101.1 |
| METRPO | -43.5 ± 3.7 | -98.5 ± 12.6 | 174.8 ± 6.2 | -29.3 ± 29.5 | -78.7 ± 5.0 | 138.5 ± 63.2 |
| TD3 | -331.6 ± 134.6 | -216.4 ± 39.6 | 168.6 ± 12.7 | -102.9 ± 101.0 | -76.5 ± 10.2 | -409.2 ± 928.8 |
| SAC | -161.6 ± 43.7 | -227.6 ± 42.2 | 159.5 ± 12.1 | -0.2 ± 0.1 | -69.4 ± 7.0 | 195.5 ± 8.7 |
| Random | -183.1 ± 41.5 | -199.0 ± 10.0 | -249.5 ± 228.4 | -205.9 ± 12.1 | -374.1 ± 15.6 | 31.3 ± 36.3 |
| Time-step | 50000 | 50000 | 50000 | 50000 | 50000 | 50000 |

Table 4: Performance of each algorithm on environments based on OpenAI Gym Brockman et al. (2016) classic control environments. In the table, we record the performance at 50000 time-step.

## A.3 FULL RESULTS OF BENCH-MARKING PERFORMANCE

In this section, we show the figures of all the environments in Figure 8. We also include the final performance in the Table 2 and 4. As we can see, POPLIN has consistently the best performance among almost all the environments. We also include the time-steps we use on each environment for all the algorithms in Table 2 and 4.

### A.3.1 HYPER-PARAMETERS

In this section, we introduce the hyper-parameters we search during the experiments. One thing to notice is that, for all of the experiments on PETS, POPLIN, we use the model type PE (probabilistic ensembles) and propagation method of E (expectation). While other combinations of model type and propagation methods might result in better performance, they are usually prohibitively computationally expensive. For example, the combination of PE-DS requires a training time of about 68 hours for one random seed, for PETS to train with 200 iteration, which is 200,000 time-step. As a matter of fact, PE-E is actually one of the best combination in many environments. Since POPLIN is based on PETS, we believe this is a fair comparison for all the algorithms.

We show the hyper-parameter search we perform for PETS in the paper in Table 5. For the hyper-parameters specific to POPLIN, we summarize them in 6 and 7.

| Hyper-parameter | Value Tried |
|---|---|
| Population Size | 100, 200, ..., 2000 |
| Planning Horizon | 30, 50, 100 |
| Initial Distribution Sigma | 0.01, 0.03, 0.1, 0.25, 0.3, 0.5 |
| CEM Iterations | 5, 8, 10, 20 |
| ELite Size $\xi$ | 50, 100, 200 |

Table 5: Hyper-parameter grid search options for PETS.

| Hyper-parameter | Value Tried |
|---|---|
| Training Data | real data, hallucination data |
| Variant | Replan, Init |
| Initial Distribution Sigma | 0.001, 0.003, 0.01, 0.03, 0.1 |

Table 6: Hyper-parameter grid search options for POPLIN-A.

| Hyper-parameter | Value Tried |
|---|---|
| Training Data | real data, hallucination data |
| Training Variant | BC, GAN, Avg |
| Noise Variant | Uni, Sep |
| Initial Distribution Sigma | 0.001, 0.003, 0.01, 0.03, 0.1 |

Table 7: Hyper-parameter grid search options for POPLIN-P. We also experiment with using WGAN in Salimans et al. (2016) to train the policy network, which does not results in good performance and is not put into the article.

### A.4 FULL RESULTS OF POLICY CONTROL

Due to the space limit, we are not able to put all of the results of policy control in the main article. More specifically, we add the figure for the original Cheetah-v0 compared to the figures shown in the main article, as can be seen in 9 (b). Again, we note that POPLIN-P-BC and POPLIN-P-GAN are comparable to each other, as mentioned in the main article. POPLIN-P-BC and POPLIN-P-GAN are the better algorithms respectively in Cheetah and Cheetah-v0, which are essentially the same environment with different observation functions.

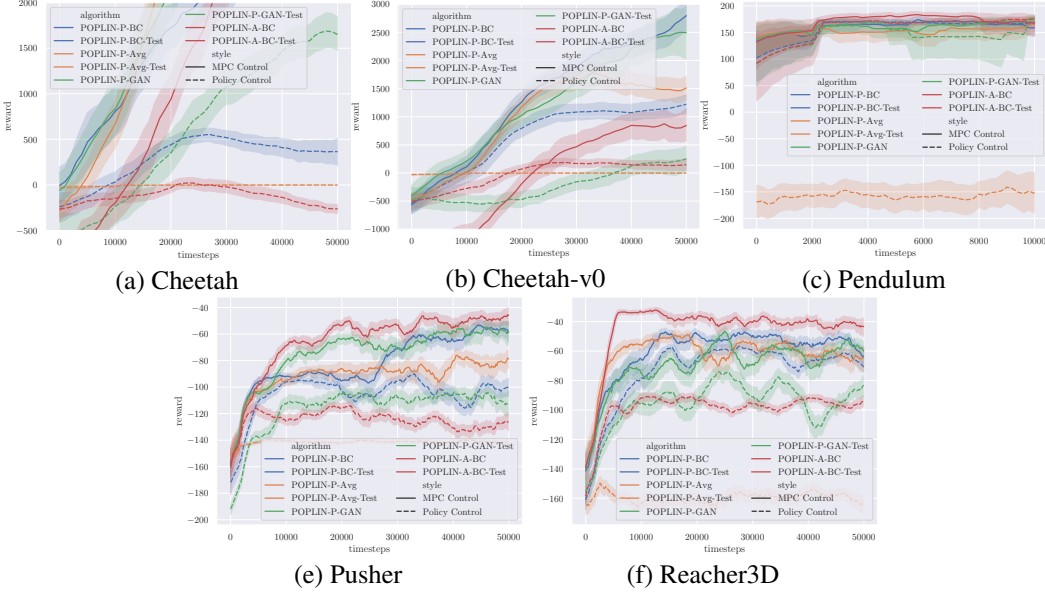

Figure 9: The planning performance and the testing performance of the proposed POPLIN-A, and POPLIN-P with its three training schemes, which are namely behavior cloning (BC), generative adversarial network training (GAN) and setting parameter average (Avg).

### A.5 ABLATION STUDY FOR DIFFERENT VARIANT OF POPLIN

In this section, we show the results of different variant of our algorithm. In Figure 11, the performances of different random seeds are visualized, where we show that POPLIN has similar randomness in performance to PETS. Additionally, we visualize POPLIN-P-BC in Figure 10 (b), whose best distribution variance for policy planning is 0.01, while the best setting for testing is 0.03.

### A.6 POPULATION SIZE

In Figure 12, we include more detailed figures of the performance of different algorithms with different population size. One interesting finding is that even with fixed parameters of zeros, POPLIN-P can still performance very efficient search. This is indicating that the efficiency in optimization of

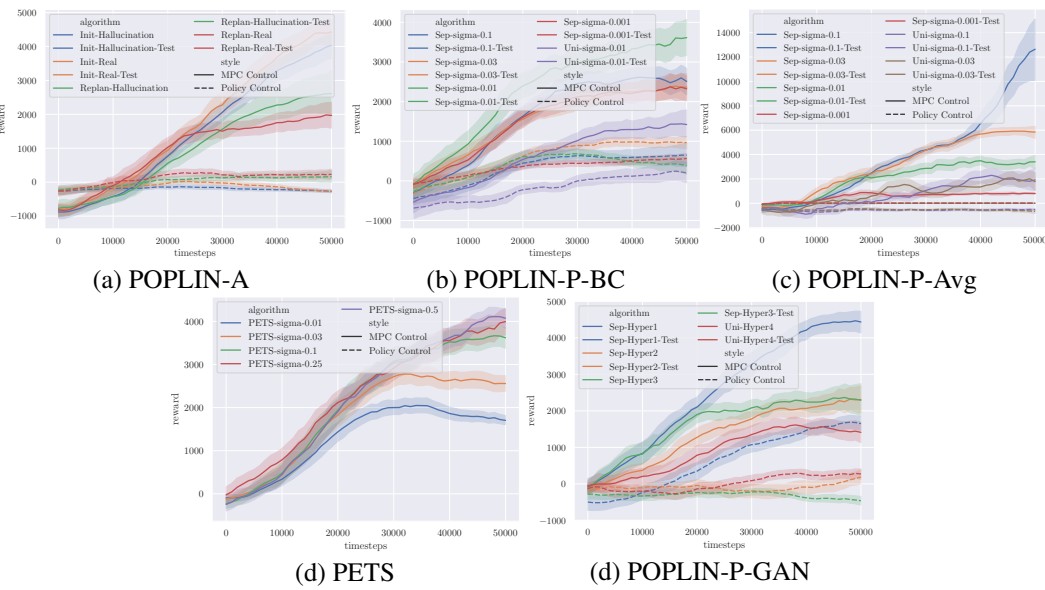

(a) POPLIN-A      (b) POPLIN-P-BC      (c) POPLIN-P-Avg

(d) PETS      (d) POPLIN-P-GAN

Figure 10: The performance of POPLIN-A, POPLIN-P-BC, POPLIN-P-Avg, POPLIN-P-GAN using different hyper-parameters. The tested environment is Cheetah.

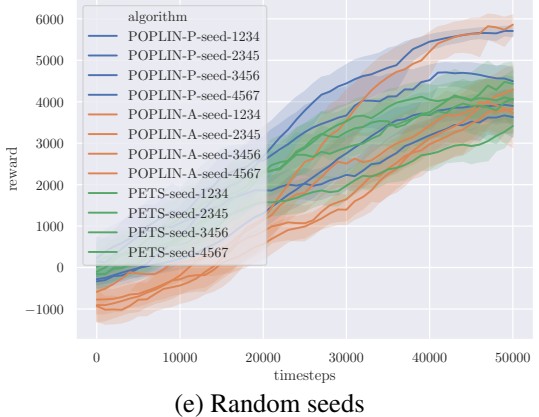

(e) Random seeds

Figure 11: The performance of POPLIN-A, POPLIN-P, and PETS of different random seeds on Cheetah environment.

POPLIN-P, especially of POPLIN-P-AVG, is the key reasons for successful planning. However, this scheme naturally sacrifices the policy distillation and thus cannot be applied without planning.

### A.7 THE REWARD SURFACE OF DIFFERENT ALGORITHM

In this section, we provide a more detailed description of the reward surface with respect the the solution space (action space for PETS and POPLIN-A, and parameter space for POPLIN-P) in Figure 13, 14, 15, 16, 17. As we can see, variants of POPLIN-A are better at searching, but the reward surface is still not smooth. POPLIN-A-Replan is more efficient in searching than POPLIN-A-Init, but the errors in dynamics limit its performance. We also include the results for POPLIN-P using a 1-layer neural network in solution space in Figure 16 (g), (h). The results indicate that the deeper the network, the better the search efficiency.

We also provide more detailed version of Figure 1 in Figure 18. We respectively show the surface for PETS, POPLIN-P-P using 1 and 0 hidden layers. Their planned trajectories across different CEM updates are visualized in Figure 19, 20, 21. Originally in Figure 1, we use the trajectories in iteration

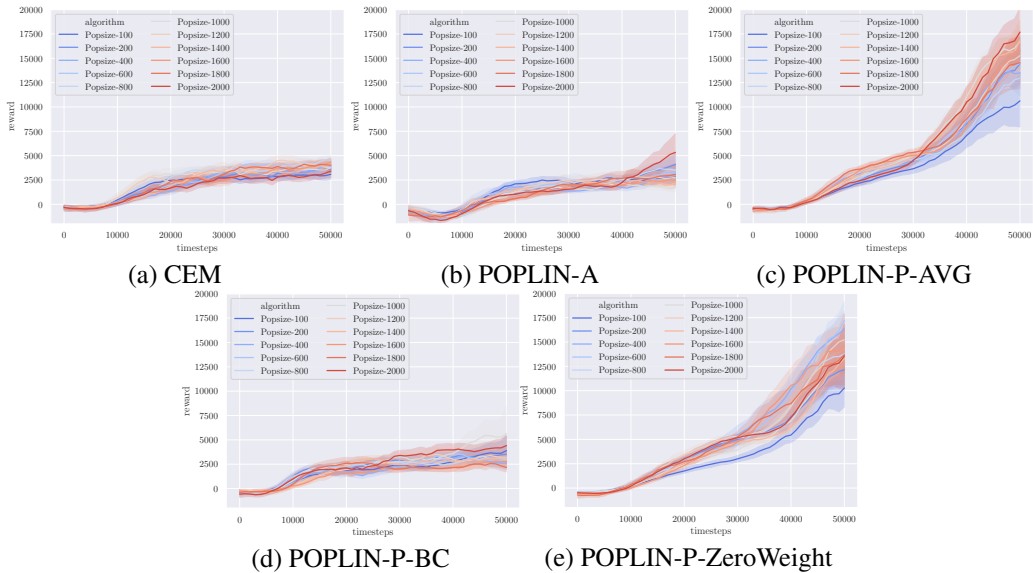

Figure 12: The performance of PETS, POPLIN-A, POPLIN-P-Avg, POPLIN-P-BC and POPLIN-P whose network has fixed parameters of zeros. The variance of the candidates trajectory $\sigma$ in POPLIN-P is set to 0.1. The tested environment is Cheetah.

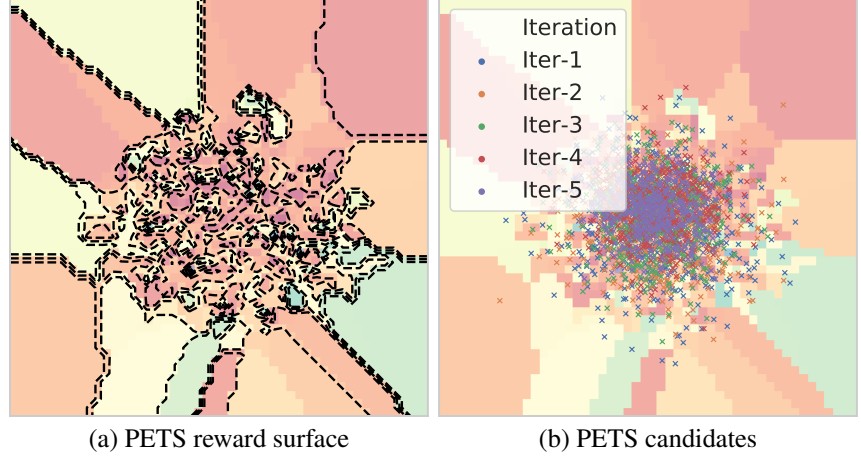

Figure 13: Reward surface in solution space (action space) for PETS algorithm.

1, 3, 5 for better illustration. In the appendix, we also provide all the iteration data. Again, the color indicates the expected cost (negative of expected reward). From left to right, we show the updated the trajectories in each iteration with blue scatters.

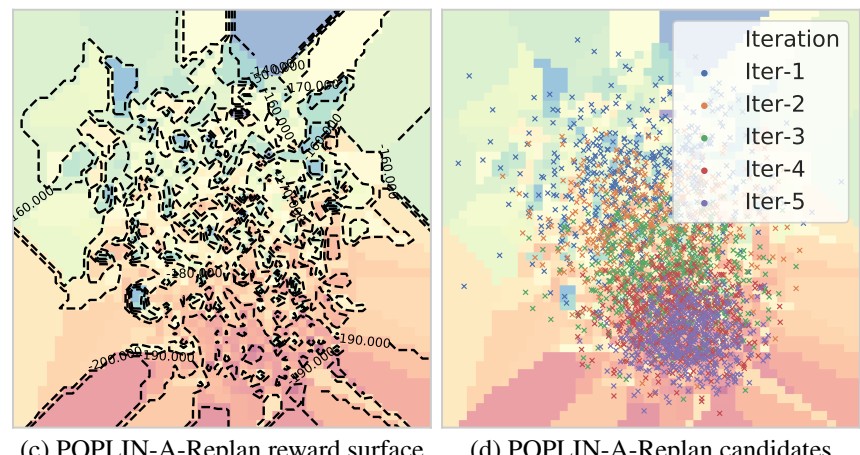

(c) POPLIN-A-Replan reward surface     (d) POPLIN-A-Replan candidates

Figure 14: Reward surface in solution space (action space) for POPLIN-A-Replan.

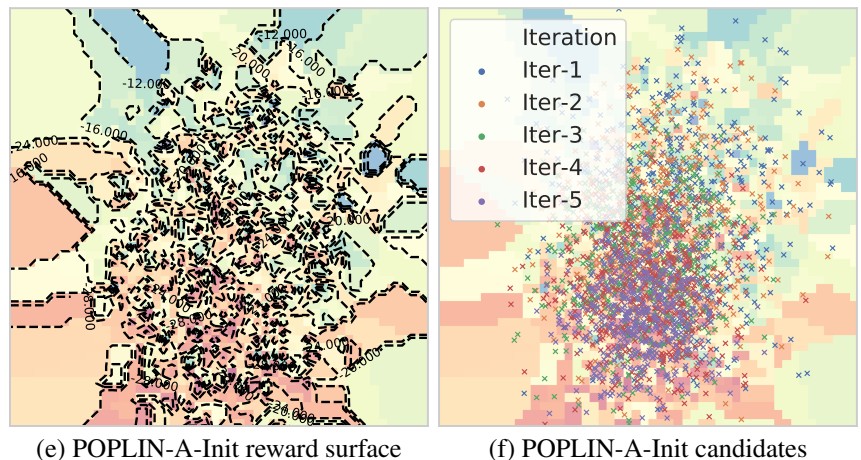

(e) POPLIN-A-Init reward surface     (f) POPLIN-A-Init candidates

Figure 15: Reward surface in solution space (action space) for POPLIN-A-Init.

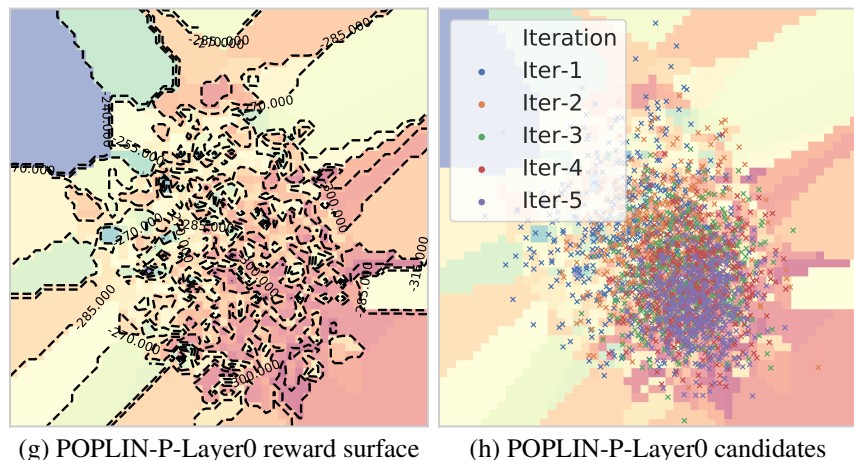

(g) POPLIN-P-Layer0 reward surface     (h) POPLIN-P-Layer0 candidates

Figure 16: Reward surface in solution space (parameter space) for POPLIN-P with 0 hidden layer.

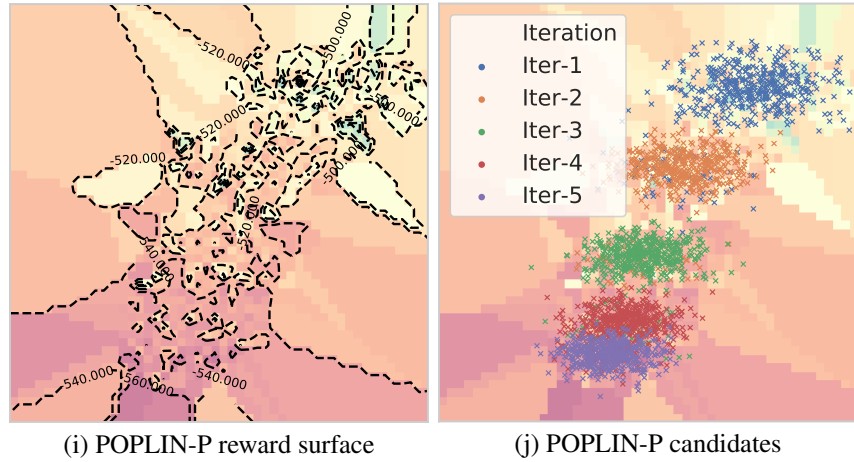

(i) POPLIN-P reward surface       (j) POPLIN-P candidates

Figure 17: Reward surface in solution space (parameter space) for POPLIN-P using 1 hidden layer.

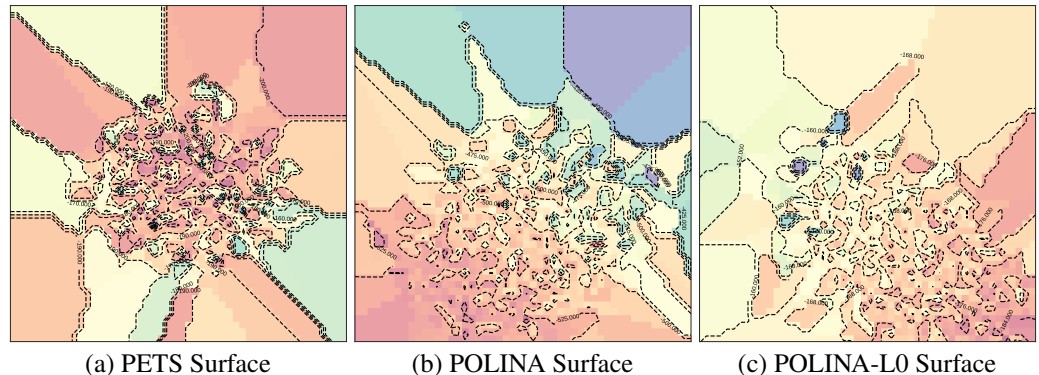

(a) PETS Surface      (b) POLINA Surface      (c) POLINA-L0 Surface

Figure 18: The color indicates the expected cost (negative of expected reward). We emphasis that all these figures are visualized in the action space. And all of them are very unsmooth. For the figures visualized in solution space, we refer to Figure 13.

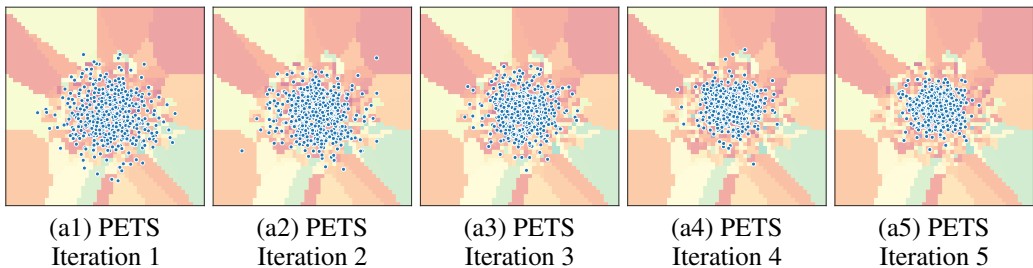

(a1) PETS Iteration 1    (a2) PETS Iteration 2    (a3) PETS Iteration 3    (a4) PETS Iteration 4    (a5) PETS Iteration 5

Figure 19: The figures are the planned trajectories of PETS.

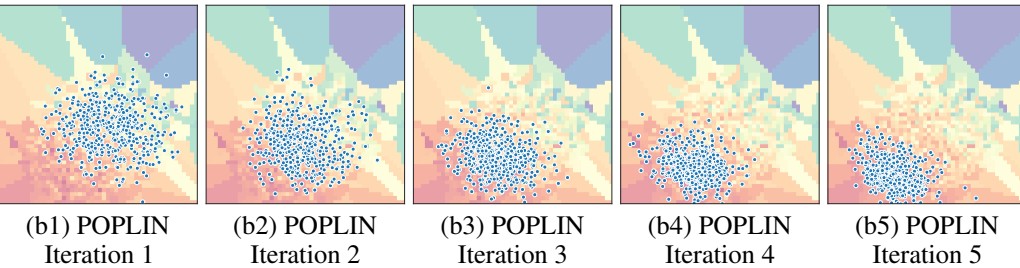

(b1) POPLIN Iteration 1    (b2) POPLIN Iteration 2    (b3) POPLIN Iteration 3    (b4) POPLIN Iteration 4    (b5) POPLIN Iteration 5

Figure 20: The figures are the planned trajectories of POPLIN-P using 1 hidden layer MLP.

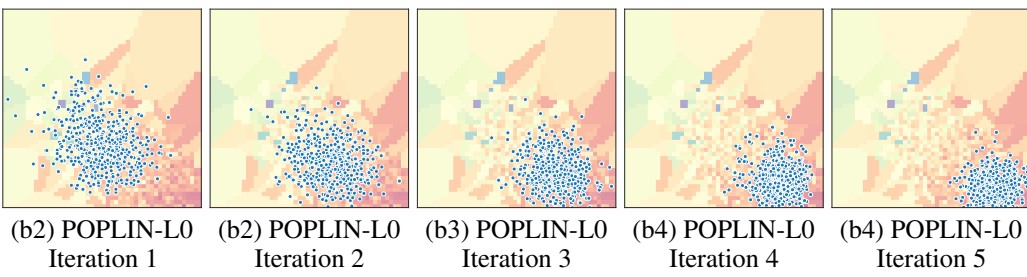

(b2) POPLIN-L0        (b2) POPLIN-L0        (b3) POPLIN-L0        (b4) POPLIN-L0        (b4) POPLIN-L0
   Iteration 1             Iteration 2             Iteration 3             Iteration 4             Iteration 5

Figure 21: The figures are the planned trajectories of POPLIN-P using 0 hidden layer MLP.

