# OpenReview forum: "Exploring Model-based Planning with Policy Networks"
_ICLR.cc/2020/Conference — Accept (Poster)_

### Official Review · AnonReviewer3 · 2019-10-19
**Official Blind Review #3**

**Rating:** 6

**Review:**

Contributions

This paper proposes a MBRL algorithm for continuous action space domains that relies on a policy network to propose an initial trajectory, which is then refined, either in the action space or in the policy space, using standard CEM.
3 options are evaluated to update the policy network: a GAN approach, a regression approach (behaviour cloning) and a direct modifications of the policy based on the perturbation found by CEM.


Review

The presentation of the method is thorough, as well as the motivations and the experiments. However, the presentation of the experimental result could gain in clarity. For example, plots in Figure 7 are totally unreadable when printed on paper, and even after zooming in the pdf it's difficult to tell what's going on. I think the text doesn't even mention on which environment these curves are obtained.
I'd suggest some normalized performance ratio over the final performance of the base algorithm, to show the contribution of all the ingredients of the method.

For the results presented in section 5.1, it's unclear to me which exact variation of POPLIN-A and POPLIN-P have been selected.


Finally the abstract reads "We show that POPLIN obtains state-of-the-art performance in the MuJoCo benchmarking environments" (similar claims are made elsewhere in the paper). I'd tone this down a little, since the said performance ist state-of-the-art only for the 200k frames regime, and as it's hinted at the end of 5.1, I assume the proposed approach suffers from the same shortcoming as other competing MBRL methods which is that their performance tends to plateau after a few hundred thousand samples. Overall, the SOTA performance in MuJoCo is held by MFRL methods, but typically require more training (up to 1M)


**Experience Assessment:**

I do not know much about this area.

**Review Assessment: Checking Correctness Of Derivations And Theory:**

I carefully checked the derivations and theory.

**Review Assessment: Checking Correctness Of Experiments:**

I assessed the sensibility of the experiments.

**Review Assessment: Thoroughness In Paper Reading:**

I read the paper at least twice and used my best judgement in assessing the paper.

---

> ### Author Response · Authors · 2019-11-15
> **Response to Reviewer 3**
>
> We thank the reviewer for the valuable suggestions, and answer the following questions raised in the helpful review.
>
> Q1. Readability of the plots in Figure 7 and missing text doesn't describing environments tested.
>
> We are sorry for the confusion. In the latest revision, we fixed the captions needed to describe the figures.
>
> Q2. I'd suggest some normalized performance ratio over the final performance of the base algorithm, to show the contribution of all the ingredients of the method.
>
> We thank the reviewer for the valuable suggestion. In the latest revision, we updated the normalized performance in the appendix.
> Due to the space limit, we summarize some of the representative results here:
>
>                      |      Cheetah     |         Ant         |       Hopper    |     Swimmer
> POPLIN-P    | 0.944±0.079    | 0.932±0.128   |  0.912±0.112  |   0.936±0.086
> POPLIN-A    | 0.395±0.057    | 0.459±0.175   |  0.582±0.175  |   0.962±0.018
> PETS            | 0.363±0.002    | 0.466±0.091   |  0.566±0.113  |   0.916±0.032
> RS                | 0.072±0.005    | 0.214±0.015   |  0.092±0.006  |   0.156±0.024
> MBMF         | 0.025±0.002    | 0.054±0.02     |  0.355±0.2      |   0.377±0.114
> TRPO           | 0.028±0.003    | 0.129±0.01     |  0.164±0.116  |   0.22±0.028
> PPO             | 0.023±0.01      | 0.128±0.02     |  0.527±0.187  |  0.489±0.037
> GPS             | 0.067±0.051    | 0.178±0.085   |  0.406±0.037  |   0.023±0.016
> METRPO      | 0.004±0.043    | 0.113±0.007   |  0.777±0.091  |   0.664±0.262
> TD3              | 0.074±0.061    | 0.348±0.114   |  0.876±0.181  |   0.28±0.327
> SAC              | 0.184±0.006    | 0.219±0.059   |  0.689±0.134  |   0.612±0.173
> Random      | 0.037±0           | 0.191±0.019   |  0.042±0.104  |   0.069±0.032
> Max, Min     |  13000, -800   |    2500, 0        |   2500, -3000  |    360, -40
>
> Q3. For the results presented in section 5.1, it's unclear to me which exact variation of POPLIN-A and POPLIN-P have been selected.
>
> Sorry for the confusion. It is also brought by other reviewers as well. We use POPLIN-A-BC-Replan and POPLIN-P-Sep respectively. We updated them in the latest revision.
>
> Q4. Overall, the SOTA performance in MuJoCo is held by MFRL methods, but typically require more training (up to 1M).
>
> Yes we agree that our algorithm as well as other MBRL algorithms plateau after a few hundred thousand samples. We rephrase our paper in the latest revision accordingly to avoid over-claiming.
> We also refer to the general response and the updated paper for more details.

---

### Official Review · AnonReviewer2 · 2019-10-21
**Official Blind Review #2**

**Rating:** 8

**Review:**

Summary

This work provides a novel model-based reinforcement learning algorithm for continuous domains (Mujoco) dubbed POPLIN. The presented algorithm is similar in vein to the state-of-the-art PETS algorithm, a planning algorithm that uses state-unconditioned action proposal distributions to identify good action sequences with CEM in the planning routine. The important difference compared to PETS is the incorporation of a parametric state-conditioned policy (trained on real data) in the planning routine to obtain better action-sequences (CEM is used to learn the "offset" from the parametric policy). The paper presents two different algorithmic ablations where CEM either operates in action space or parameter space (POPLIN-A and POPLIN-P respectively), in combination with different objectives to learn the parametric policy. The method is evaluated on 12 continuous benchmarks and compared against state-of-the-art model-based and model-free algorithms, indicating dominance of the newly proposed method.

Quality

The quality of the paper is high. This is an experimental study and the number of benchmarks and baselines is far above average compared to other papers in that field. One minor point is that averaging experiments over 4 seeds only is usually not optimal in these environments, but in light of the sheer amount of baselines and benchmarks excusable. While the experimental results are impressive, the authors mention that asymptotic performance in Walker2d and Humanoid might not match the asymptotic performance of model-free baselines. This could be stated more clearly. Also, there are no Humanoid experiments in the paper despite mentioned in the text (2nd paragraph in Section 5.1)?

Clarity

The clarity of the paper can be in parts improved upon. For example, how does the "policy control" ablation (mentioned in Section 4.3) work precisely, i.e. the interplay between executing the parametric policy in the real world and harnessing the environment model? I assume the policy distillation techniques in Section 4.4 are different alternatives for the second-to-last lines in the pseudocodes from the appendix? Which one is the default used in Section 5.1? On a minor note, above Equation (7), a target network is mentioned---where does the target network occur in Equation (7)? There are some plots that do not mention the name of the environment, e.g. in Figure (4), but also some in the appendix. Furthermore, it could be stated more clearly that the reward function is assumed to be known. If the authors improve the clarity of their paper significantly, I am willing to increase my score further (presupposing that no severe issues arise in the discussion phase).

Originality

Adding a parametric policy to PETS is not the most original idea, but clearly a gap in the current literature.

Significance

The experiments and the empirical results make the paper quite significant.

Update

The authors improved the clarity of the paper. I therefore increase to 8. Section 4.4 paragraph "Setting parameter average (AVG)" can still be improved---does this go together with POPLIN-P-Uni from Section 4.2?

**Experience Assessment:**

I have published one or two papers in this area.

**Review Assessment: Checking Correctness Of Derivations And Theory:**

I assessed the sensibility of the derivations and theory.

**Review Assessment: Checking Correctness Of Experiments:**

I assessed the sensibility of the experiments.

**Review Assessment: Thoroughness In Paper Reading:**

I read the paper at least twice and used my best judgement in assessing the paper.

---

> ### Author Response · Authors · 2019-11-15
> **Response to Reviewer 2. We hope that the current revision has greatly improved its clarity**
>
> We thank the reviewer for the valuable suggestion. In the latest revision, we improve the clarity of the paper accordingly and apologize again for the confusion caused by our writing in the original revision.
>
> Q1. Walker2d and Humanoid might not match the asymptotic performance of model-free baselines. This could be stated more clearly.
>
> We thank Reviewer 2 for the suggestion. We agree that it should be stated more clearly, and we updated in the newest revision.
>
> Q2. Also, there are no Humanoid experiments in the paper despite mentioned in the text (2nd paragraph in Section 5.1)?
>
> Sorry for the confusion. In section 5.1, we were meant to mention humanoid as the tasks which POPLIN cannot solve efficiently. Therefore we didn’t put the results in the paper.
> We rephrase the sentences to remove ambiguity in the latest revision.
>
> Q3. how does the "policy control" ablation (mentioned in Section 4.3) work precisely?
>
> We apologize for the confusion of “policy control” and “MPC control” in our paper.
> “MPC Control”: To generate the data that is used to train the policy network, we always use “MPC planning”, which uses the distilled policy network, learnt dynamics and plans into the future at every time-steps.
> “Policy Control”: We are also interesting to see how powerful the distilled policy network along is, without the use of expensive planning with learnt dynamics.
> To get the “policy control” performance, we load the saved policy network weights at different iterations. The data generated is not used in “MPC control” or training.
>
> Q4. Policy distillation techniques in Section 4.4 and pseudocodes.
>
> Yes, policy distillation techniques are in Section 4.4 are different alternatives for the second-to-last lines in the pseudocodes. For the benchmarking results in Section 5.1, POPLIN-A-BC-Replan and POPLIN-P-Sep-AVG variants are respectively used in the paper. We also provide the results of the  ablation study of different distillation techniques in Section 5.4.
>
> Q5. On a minor note, above Equation (7), a target network is mentioned---where does the target network occur in Equation (7)?
>
> It should have been just “network” instead of “target network”. The “target” was redundant and clearly caused confusion. We were meant to indicate we don’t have noise randomly generated during training.
>
> Q6. There are some plots that do not mention the name of the environment, e.g. in Figure (4), but also some in the appendix.
>
> We thank the reviewer for pointing out. They are fixed in the latest revision.
>
> Q7. Furthermore, it could be stated more clearly that the reward function is assumed to be known.
>
> In the latest revision, we now state more clearly about the reward function.
>
> Q8. If the authors improve the clarity of their paper significantly, I am willing to increase my score further.
>
> We apologize again for our lack of clarity. We thank the reviewer to the valuable suggestions.
> We uploaded the updated paper in the openreview system, which we hope has greatly improved the readability of the paper.

---

### Official Review · AnonReviewer1 · 2019-10-23
**Official Blind Review #1**

**Rating:** 6

**Review:**

This paper presents POPLIN, a novel model-based reinforcement learning algorithm, which trains a policy network to improve model-prediction control. The paper studies extensively how to utilize the policy, by planning in action space or planning in parameter space and how to train the policy, by behavioral cloning, by GAN or by averaging the results of CEM. The experiments show that the proposed algorithm can perform very well in MuJoCo tasks.

Overall, the paper is well-written and the method is novel. The extensive experiments make the paper more convincing.

Questions:
1. In the example of arm in the first paragraph of Section 4.2, although the mean is 0, the randomness in sampling will be the tie breaker. So "failing" is probably not the best word here.
2. It seems that the policy network is deterministic. Why?
3. Reparametrizable policy often requires fewer (noise) parameters to be optimized over. For example, suppose the policy outputs a multi-variate Gaussian distribution with diagonal covariance in R^10, then we only need to optimize over 10 parameters (the Gaussian noise).  Why optimize all parameters in the policy network, which makes optimization more difficult?
4. Sec 5.1 Para 2: To my knowledge, the state-of-the-art model-based RL algorithm in MuJoCo environments is MBPO (https://arxiv.org/abs/1906.08253, NeurIPS 2019).
5. What's the architecture of policy network? More importantly, how many parameters does the policy network have? It's really interesting to see that CEM works for such a high dimensional space. In an environment where a larger network is required, the optimization seems to be more difficult.
6. In Ablation Study, what does "imaginary data" mean?
7. I'm also curious to see how the objective of CEM improves.

Minor comments:

1. Sec 5.1 Para 2 L11: efficient -> efficiently.
2. Are you talking about Figure 3 at Sec 5.2? If so, could you please add a link to the figure?
3. A lot of default values need to be specified: What's the policy distillation method used in POPLIN-A/P in Table 1? Does POPLIN-A mean POPLIN-A-Replan or POPLIN-A-init? Does POPLIN-P mean POPLIN-P-Sep or POPLIN-P-Uni?
4. Sec 4.1 Eq (2): \delta_0...\delta_\xi are \xi+1 sequences.
5. Sec 5.3 Para 3: multi-model -> multi-modal.

**Experience Assessment:**

I have published one or two papers in this area.

**Review Assessment: Checking Correctness Of Derivations And Theory:**

I assessed the sensibility of the derivations and theory.

**Review Assessment: Checking Correctness Of Experiments:**

I carefully checked the experiments.

**Review Assessment: Thoroughness In Paper Reading:**

I read the paper at least twice and used my best judgement in assessing the paper.

---

> ### Author Response · Authors · 2019-11-15
> **Response to Reviewer 1**
>
> We thank the reviewer for the suggestion and updated accordingly in the latest revision.
>
> Q1. In the example of arm in the first paragraph of Section 4.2, although the mean is 0, the randomness in sampling will be the tie breaker. So "failing" is probably not the best word here.
>
> We agree that “failing” is not the most appropriate word. In the newest version, we rephrase it into “it is hard to model the bimodal action distribution in traditional planner”.
>
>
> Q2. Why use deterministic policy network?
>
> In CEM, the stochasticity is partially modeled by the planner by maintaining a population of candidates.
> One of the reasons we didn’t have an explicit stochastic policy network (which sample actions from a distribution) is that it might make the CEM updates unstable and hard to converge.
> In the paper we didn’t test the idea, but we think stochastic policy network is definitely worth a try in POPLIN.
>
> Q3. Why optimize all parameters in the policy network, which makes optimization more difficult?
>
> The case where we only optimize the last layer (covariance) of the policy can be regarded as a special case of POPLIN-A.
> We believe there’s a trade-off between capacity and optimization difficulty.
> The more parameters or the deeper the network, the more expressive but also more difficult to optimize the planner can be.
> Empirically, we found that for low dimensional environments, it does turn out having more parameters does not help the planner, and only brings more computation cost. We also refer to Q5 below for more details.
>
> Q4. Comparison to MBPO.
>
> We thank the reviewer for bringing this baseline. We added it into the related work section in the last revision and modified the claim in the paper.
> MBPO and POPLIN were developed with very different techniques (Dyna for MBPO and MPC for POPLIN), and the environments / reward functions are defined differently.
> We have similar performance in many tasks, and MBPO seems to be very effective for Ant and Walker. We believe the performance will be even better if we combine the Dyna updates from MBPO and the MPC in POPLIN.
>
> Q5. What's the architecture of policy network? More importantly, how many parameters does the policy network have?
>
> We experimented with different sizes of MLP, including [32] (best for POPLIN-P), [64], [32, 32] (best for POPLIN-A), and [64, 64]. Very deep network is both very computationally infeasible and hard to optimize.
>
> Q6. In Ablation Study, what does "imaginary data" mean?
>
> Sorry for the confusion. The imaginary data refers to the planned trajectories generated in rollouts. They are predicted by the learnt dynamics, and can potentially helpful to train the policy network.
>
>
> Q8. A lot of default values need to be specified: What's the policy distillation method used in POPLIN-A/P in Table 1? Does POPLIN-A mean POPLIN-A-Replan or POPLIN-A-init? Does POPLIN-P mean POPLIN-P-Sep or POPLIN-P-Uni?
>
> Sorry for the confusion. We use POPLIN-A-BC-Replan and POPLIN-P-Sep respectively. We updated them in the latest revision.

---

### Author Response · Authors · 2019-11-15
**General response and update of the paper**

We thank the reviewers for the valuable suggestions and acknowledgement of the paper.
We updated our paper and summarize the revisions as follows:

1. We greatly improve the clarity of the paper.

We are sorry for the confusion caused in our earlier version of the paper, which, for example, didn’t specify the model variants used in the experiment section, and has inconsistent use of words.

2. We added more papers in the related work section.

Some of the concurrent or latest research are now included in the paper.

We also refer to the responses to each reviewer, and we are hoping for further feedback.

---

### Decision · Program_Chairs · 2019-12-19

**Decision:**

Accept (Poster)

**Comment:**

This paper proposes a model-based policy optimization approach that uses both a policy and model to plan online at test time. The paper includes significant contributions and strong results in comparison to a number of prior works, and is quite relevant to the ICLR community. There are a couple of related works that are missing [1,2] that combine learned policies and learned models, but generally the discussion of prior work is thorough. Overall, the paper is clearly above the bar for acceptance.

[1] https://arxiv.org/pdf/1703.04070.pdf
[2] https://arxiv.org/pdf/1904.05538.pdf